# ConStruct: Structural Distillation of Foundation Models for Prototype-Based Weakly Supervised Histopathology Segmentation

**Khang Le**[*1]      KHANG.LEKHANGLE23@HMCUT.EDU.VN

**Ha Thach**[*2]      NGUYENBICHHA.THACH@STUDENT.UTS.EDU.AU

**Anh M. Vu**[*3]      MVU9@COUGARNET.UH.EDU

**Trang T. K. Vo**[4]      TRANGVTK.18@GRAD.UIT.EDU.VN

**Han H. Huynh**[5]      M658112001@TMU.EDU.TW

**David Yang**[6]      DONGJUN.YANG@EMORY.EDU

**Minh H. N. Le**[7]      JOHNMINHLE@IEEE.ORG

**Thanh-Huy Nguyen**[8]      THANHHUN@ANDREW.CMU.EDU

**Akash Awasthi**[3]      AAWASTH3@CougarNet.UH.EDU

**Chandra Mohan**[3]      CMOHAN@Central.UH.EDU

**Zhu Han**[3]      ZHAN2@Central.UH.EDU

**Hien Van Nguyen**[3]      HVNGUY35@Central.UH.EDU

[1] *Ho Chi Minh City University of Technology, Vietnam*

[2] *University of Technology Sydney, Australia*

[3] *University of Houston, Houston, TX, USA*

[4] *University of Information Technology, Ho Chi Minh City, Vietnam*

[5] *College of Medical Science and Technology, Taipei Medical University, Taipei, Taiwan*

[6] *Department of Computer Science, Emory University, Atlanta, GA, USA*

[7] *Montefiore Medical Center, Albert Einstein College of Medicine, Bronx, NY, USA*

[8] *School of Computer Science, Carnegie Mellon University, Pittsburgh, PA, USA*

**Editors:** Accepted for publication at MIDL 2026

## Abstract

Weakly supervised semantic segmentation (WSSS) in histopathology relies heavily on classification backbones, yet these models often localize only the most discriminative regions and struggle to capture the full spatial extent of tissue structures. Vision–language models such as CONCH offer rich semantic alignment and morphology-aware representations, while modern segmentation backbones like SegFormer preserve fine-grained spatial cues. However, combining these complementary strengths remains challenging, especially under weak supervision and without dense annotations. We propose a prototype learning framework for WSSS in histopathological images that integrates morphology-aware representations from CONCH, multi-scale structural cues from SegFormer, and text-guided semantic alignment

---

[*] Contributed equally

to produce prototypes that are simultaneously semantically discriminative and spatially coherent. To effectively leverage these heterogeneous sources, we introduce text-guided prototype initialization that incorporates pathology descriptions to generate more complete and semantically accurate pseudo-masks. A structural distillation mechanism transfers spatial knowledge from SegFormer to preserve fine-grained morphological patterns and local tissue boundaries during prototype learning. Our approach produces high-quality pseudo masks without pixel-level annotations, improves localization completeness, and enhances semantic consistency across tissue types. Experiments on BCSS-WSSS datasets demonstrate that our prototype learning framework outperforms existing WSSS methods while remaining computationally efficient through frozen foundation model backbones and lightweight trainable adapters. Code is available at https://github.com/tom1209-netizen/ConStruct.

**Keywords:** Weakly Supervised Semantic Segmentation, Histopathology Image Analysis

## 1. Introduction

Histopathology plays a central role in cancer diagnosis by enabling the microscopic examination of tissue morphology, cellular organization, and pathological alterations. With the increasing shift toward computational and digital pathology, digitized histopathological images have become essential for supporting computer-aided diagnosis, prognosis, research, and education (Vorontsov et al., 2024; Kang et al., 2025). Semantic segmentation in histopathological images aims to delineate different types of tissue and structures. It is an important problem as it approximates the manual workflow of expert pathologists and improves both diagnostic accuracy and efficiency (Kang et al., 2025). Despite these advantages, histopathological image segmentation remains constrained by several practical challenges, including the scarcity of pixel-level annotations, the reliance on highly specialized domain experts, the labor-intensive and time-consuming nature of manual labeling, and the substantial storage and computational requirements for gigapixel whole-slide images (Zhu et al., 2023). To mitigate the lack of dense pixel-level annotations, weakly supervised semantic segmentation (WSSS) in histopathological images has been proposed to leverage less precise annotation types such as image-level labels, bounding boxes, point annotations or scribbles, arranged by ascending annotation cost (Zhu et al., 2023). Among these, image-level labels provide the weakest supervision because they lack spatial information about class locations (Zhu et al., 2023).

A typical image-level WSSS pipeline first trains a classification backbone to extract discriminative features or activation maps for pseudo-mask generation, followed by a segmentation model trained under the supervision of these pseudo-masks. The choice between CNN-based and transformer-based classification architectures is largely determined by the pseudo-mask generation strategy and the subsequent refinement modules. Pseudo-mask generation methods in histopathological WSSS in can be broadly categorized into MIL-based (Xu et al., 2019; Li et al., 2023; He et al., 2024b), CAM-based (Han et al., 2021b; Zhang et al., 2023), and prototype-based (Tang et al., 2025; Le et al., 2025) approaches. MIL-based approaches treat each histopathological image as a bag and its constituent regions (pixels or patches) as instances. Similar to the limitations observed in MIL in WSI, MIL-based methods in histopathology images also often suffer from spatial imprecision, since pixel-level predictions emerge indirectly from image-level supervision. This indirect supervision makes the generated pseudo-masks noisy, inconsistent, and unable to reliably

capture fine-grained boundaries (He et al., 2024a). Several studies had attempted to address these limitations. SA-MIL integrated a self-attention module to capture long-range spatial dependencies and reduces instance-level ambiguity and produces more coherent, boundary-aware pseudo-masks (Li et al., 2023). He et al. (He et al., 2024b) introduced a framework that leverages knowledge distillation on MIL-based pseudo-labels, where an iterative teacher–student process progressively refines MIL-derived segmentation maps through repeated denoising and representation alignment. However, despite these improvements, the inherent weakness of MIL motivates the need for spatially grounded pseudo-mask generation strategies; CAM-based methods have became the most widely used because they generate pseudo-masks directly from classification backbones (Zhu et al., 2023). However, CAMs inherently focus on the most discriminative regions and do not fully capture tissue patterns, especially in histopathology images, which have challenges in intra-class heterogeneity and inter-class homogeneity (Tang et al., 2025; Zhang et al., 2023; Wu and Zhang, 2025). Intra-class heterogeneity refers to substantial visual variation within the same tissue class, caused by differences in staining, morphology, texture, or tumor grade, while inter-class homogeneity describes the phenomenon where different tissue classes exhibit highly similar visual patterns, making them difficult to distinguish (Tang et al., 2025). Recent interpretability work, such as CIG (Vu et al., 2025), further demonstrates that, compared with other attribution methods, gradient-based CAMs capture only partial class-discriminative evidence, underscoring their limitations as the sole source of pseudo-masks.

Prototype-based methods have recently emerged as a promising alternative by modeling class-specific morphological patterns through learnable prototypes (Le et al., 2025; Sacha et al., 2025) or clustering-based prototypes (Tang et al., 2025; Pan et al., 2023). Prototypes are representative feature vectors in an embedding space that capture characteristic patterns of each semantic class (Zhou et al., 2022). They serve as reference points for classifying data samples based on proximity or similarity. This enables improved region consistency, better coverage of tissue patterns, and enhanced interpretability (Zhou et al., 2022). However, prototype learning in histopathological WSSS remains challenging. Non-learnable prototype methods typically employ clustering algorithms such as k-means (Tang et al., 2025; Pan et al., 2023) to identify representative features from patch embeddings. Although conceptually straightforward, they require computing similarity between millions of patch-level features, leading to substantial memory and computational cost. On the other hand, learnable prototype approaches address computational concerns by treating prototypes as trainable parameters optimized with a feature encoder. LPD (Le et al., 2025) is a learnable prototype–based WSSS framework in which class-specific prototypes are trained end-to-end to produce prototype-derived CAMs, refined into pseudo-masks through contrastive foreground/background alignment, while a diversity regularizer ensures that different prototypes within the same class capture distinct tissue patterns. However, learnable prototypes are highly dependent on the quality of the encoder's representation.

Different backbone architectures or foundation models exhibit distinct strengths and limitations when applied to histopathological images (Zhou et al., 2022). Although CNN-based encoders (VGG (Chan et al., 2019), ResNet (Han et al., 2021b)) excel at capturing strong local textural patterns and precise spatial localization, which is crucial for delineating tissue boundaries, their limited receptive fields constrain global context modeling, making it difficult to distinguish tissue classes that differ subtly across large spatial re-

gions. In contrast, ViT-based encoders provide rich global semantic understanding through self-attention mechanisms, beneficial for class-level discrimination. However, their token embeddings often suffer from oversmoothing and loss of spatial locality due to global attention (Ru et al., 2023; Tang et al., 2025), causing prototypes derived from ViT features to be semantically strong but spatially unstable, resulting in blurred segmentation boundaries. SegFormer addresses this partially by combining hierarchical transformer features with lightweight MLP decoders, offering strong multi-scale representations that bridge local and global context (Tang et al., 2025; Le et al., 2025).

Beyond general-purpose encoders, histopathology-specific foundation models such as CONCH (Lu et al., 2024), UNI (Chen et al., 2024), Prov-GigaPath (Xu et al., 2024) produce morphology-aware representations by pretraining on millions of tissue patches. These representations capture domain-relevant patterns that conventional encoders may fail to learn. However, these pathology foundation models lack the multi-scale structural detail required for fine-grained segmentation. Meanwhile, recent advances in vision–language models (CLIP (Radford et al., 2021), CONCH (Lu et al., 2024)) show that text-guided representations can introduce explicit semantic supervision that helps disambiguate visually similar yet conceptually distinct tissue patterns, a valuable capability in histopathology, where subtle morphological differences often carry diagnostic significance. This motivates the question of how we can design a prototype learning framework that yields prototypes that are both semantically discriminative and spatially coherent, enabling accurate and reliable pseudo-mask generation for weakly supervised histopathological segmentation, leveraging text-guided semantic alignment on morphology-aware features.

Therefore, we propose a unified prototype learning framework for WSSS in histopathological images by integrating morphology-aware representations from CONCH, multi-scale structural cues from SegFormer, and text-guided semantic alignment to produce prototypes that are simultaneously semantically discriminative and spatially coherent. Our contributions in this work are as follows:

- We propose a prototype-based pipeline that leverages text-guided initialization to address the coverage limitations of standard Class Activation Maps (CAMs). By incorporating pathology descriptions, our approach aims to generate more complete and semantically accurate pseudo-masks.

- We incorporate a structural distillation mechanism that transfers spatial knowledge from a SegFormer teacher to the student model, assisting in verifying and preserving local tissue boundaries.

- We demonstrate robust performance on histopathology benchmarks while maintaining high parameter efficiency. Our framework freezes the foundation model backbones and only trains lightweight adapters, effectively reducing learnable parameters without compromising segmentation quality.

## 2. Method

**Overview.** We propose a teacher-student distillation framework built on text-guided learnable prototypes for weakly supervised semantic segmentation (WSSS) in histopathology.

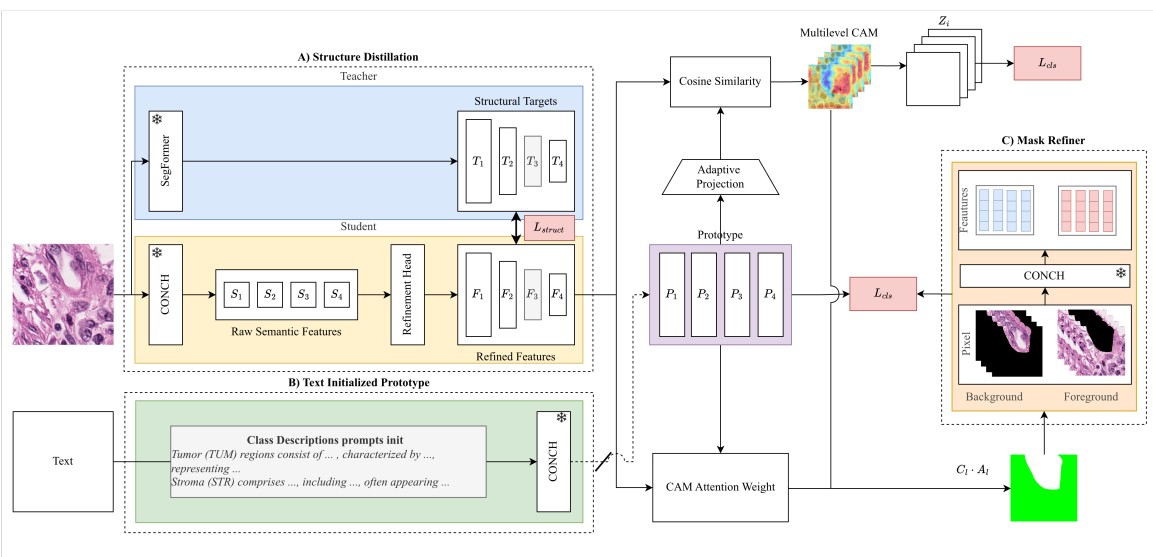

Figure 1: Overview of the proposed three-component framework. **(a) Structural Knowledge Distillation:** Frozen CONCH and SegFormer encoders extract multi-scale features; lightweight adapters refine CONCH features via relational distillation from SegFormer. **(b) Text Initialized Prototype:** Text descriptions initialize learnable prototypes **(c) Mask Refinement:** Foreground background contrastive CAM refinement

Our method comprises three main components: (1) *Structural Knowledge Distillation*, which transfers multi-scale structural priors from a frozen SegFormer teacher to lightweight trainable adapters operating on frozen CONCH features; (2) *Text-Guided CAM Generation*, which uses pathology text descriptions to initialize learnable class prototypes that produce activation maps via cosine similarity with refined visual features; and (3) *Mask Refinement*, an inference-only module that combines foreground/background contrastive alignment with Dense CRF to refine coarse CAMs into precise pixel-level predictions. The overall pipeline is illustrated in Fig. 1.

## 2.1. Structural Knowledge Distillation

**Dual encoders and multi-scale features.** Given an input image $\mathbf{x} \in \mathbb{R}^{3 \times H \times W}$, we extract four-scale feature pyramids from: (1) a frozen SegFormer MiT-B1 (Xie et al., 2021) teacher providing structural guidance, and (2) a frozen CONCH ViT-B/16 (Lu et al., 2024) student providing semantic representations.

The student produces multi-scale feature maps by extracting intermediate ViT block activations at layers 3, 6, 9, and 12, which are reshaped and interpolated to form :

$$\mathbf{F}_k^s = f_{\text{CONCH}}^{(k)}(\mathbf{x}) \text{ where } \mathbf{F}_k^s \in \mathbb{R}^{B \times C_s \times H_k \times W_k}$$

and the teacher outputs structural feature maps: $\mathbf{F}_k^t = f_{\text{SegFormer}}^{(k)}(\mathbf{x}), \mathbf{F}_k^t \in \mathbb{R}^{B \times C_t \times H_k \times W_k}$

**Lightweight adapters.** Since all backbone parameters are frozen, we introduce trainable adapters to refine CONCH features. Each feature $\mathbf{F}_k^s$ is resized and processed through a lightweight residual module: $\widetilde{\mathbf{F}}_k^s = \text{Resize}(\mathbf{F}_k^s), \mathbf{R}_k = \text{Adapter}_k(\widetilde{\mathbf{F}}_k^s)$, where the adapter follows a $1 \times 1 \to 3 \times 3(\text{dw}) \to 1 \times 1$ bottleneck design:

$$\text{Adapter}_k(\mathbf{F}) = \phi_3(\text{GELU}(\text{BN}[\phi_2(\text{GELU}(\text{BN}(\phi_1(\mathbf{F}))))]))$$

The refined features $\mathbf{R}_k$ serve as the student representation used for distillation and CAM generation. This design introduces only 6.3M trainable parameters (3.7% of total), keeping computational overhead minimal.

**Relational structural distillation.** Instead of aligning individual feature vectors, we align *pairwise token relations* to transfer boundary-aware structural knowledge. Teacher and student features are reshaped into token sequences and $\ell_2$-normalized:

$$\mathbf{S}_k = \text{Norm}(\text{Reshape}(\mathbf{R}_k)) \quad \text{and} \quad \mathbf{T}_k = \text{Norm}(\text{Reshape}(\text{Resize}(\mathbf{F}_k^t)))$$

Affinity matrices encode token-to-token similarity: $A_k^{\text{stu}} = \mathbf{S}_k \mathbf{S}_k^\top$ and $A_k^{\text{tea}} = \mathbf{T}_k \mathbf{T}_k^\top$. The relational distillation objective averages an MSE loss across selected guidance layers $\mathcal{K}$ (we use layer 2 by default):

$$\mathcal{L}_{\text{struct}} = \frac{1}{|\mathcal{K}|} \sum_{k \in \mathcal{K}} \text{MSE}(A_k^{\text{stu}}, A_k^{\text{tea}})$$

This transfers boundary-aware structural priors with extremely low computational overhead, since only adapters are updated.

## 2.2. Text-Guided Prototype CAMs

**Text-initialized prototypes.** For each semantic class $c \in \{1, \ldots, C\}$, we provide a detailed pathology description (Table 1) and encode it using the frozen CONCH text tower: $\mathbf{t}_c = f_{\text{text}}(\text{prompt}_c)$. A shared two-layer MLP $\phi_{\text{proj}}$ with hidden dimension $D_{\text{hidden}} = n_{\text{ratio}} \times D_{\text{proto}}$ projects text embeddings to the visual feature dimension: $\mathbf{p}_c' = \phi_{\text{proj}}(\mathbf{t}_c), \tilde{\mathbf{p}}_c = \text{Norm}(\mathbf{p}_c')$, forming a learnable prototype bank:

$$\tilde{\mathbf{P}} = [\tilde{\mathbf{p}}_1, \ldots, \tilde{\mathbf{p}}_C]^\top \in \mathbb{R}^{C \times D_{\text{proto}}}$$

**Adaptive projection to feature space.** To align prototypes with the final refined features $\mathbf{R}_4$, we apply an *Adaptive Layer* (two-layer MLP) that projects prototypes from dimension $D_{\text{proto}}$ to $C_s$: $\mathbf{P}_4 = \text{AdaptiveLayer}(\tilde{\mathbf{P}}) \in \mathbb{R}^{C \times C_s}$.

**Prototype-to-feature similarity and CAMs.** Refined student features are converted to normalized tokens: $\hat{\mathbf{R}}_4 = \text{Norm}(\text{Reshape}(\mathbf{R}_4))$. Cosine similarity with prototypes yields class activation maps $\mathbf{G}_4 = \tau \cdot (\hat{\mathbf{R}}_4 \mathbf{P}_4^\top)$, where $\tau$ is a learnable logit scaling factor initialized to $1/0.07$. Image-level supervision is applied through global average pooling: $\mathbf{z} = \text{GAP}(\mathbf{G}_4)$ with $\mathcal{L}_{\text{cls}} = \text{BCEWithLogits}(\mathbf{z}, \mathbf{y})$.

## 2.3. Mask Refinement

**Adaptive thresholding and pseudo masks.** Following PBIP (Tang et al., 2025), we generate pseudo masks by adaptive thresholding: $t = \alpha \max_{\mathbf{x}} \text{cam}(\mathbf{x})$ with $\alpha = 0.5$,

and define foreground and background indicator functions: $\mathbb{1}_{\text{fg}}(\mathbf{x}) = \mathbf{1}\{\text{cam}(\mathbf{x}) \geq t\}$ and $\mathbb{1}_{\text{bg}}(\mathbf{x}) = 1 - \mathbb{1}_{\text{fg}}(\mathbf{x})$.

**Foreground/background contrastive alignment.** We crop masked FG and BG regions and encode their features using a frozen CONCH (Lu et al., 2024) image encoder, obtaining region embeddings $\mathbf{f}_{\text{fg}}$ and $\mathbf{f}_{\text{bg}}$. We employ InfoNCE-style contrastive losses with a memory bank of 2048 negative samples:

- **FG loss** $\ell_{\text{fg}}$: FG features are pulled toward prototypes of the same class and pushed away from prototypes of other classes and memory bank negatives.
- **BG loss** $\ell_{\text{bg}}$: BG features are attracted to background prototypes and repelled from all foreground prototypes.

The refiner loss combines both terms: $\mathcal{L}_{\text{sim}} = \ell_{\text{fg}} + \ell_{\text{bg}}$.

**Total training objective.** The final loss combines classification, structural distillation, and contrastive alignment: $\mathcal{L}_{\text{total}} = \lambda_{\text{cls}}\mathcal{L}_{\text{cls}} + \lambda_{\text{struct}}\mathcal{L}_{\text{struct}} + \lambda_{\text{sim}}\mathcal{L}_{\text{sim}}$, with weights $\lambda_{\text{cls}} = 1.0$, $\lambda_{\text{struct}} = 1.5$, and $\lambda_{\text{sim}} = 0.2$.

**Inference refinement.** During inference, we apply test-time augmentation (6 augmentations: horizontal flip $\times$ brightness scaling $\{0.9, 1.0, 1.1\}$) and average the resulting CAMs. Background probability is estimated as $(1 - \max_c M_c)^{10}$.

**Dense Conditional Random Field.** To further refine object boundaries and impose spatial consistency, we apply a fully connected Conditional Random Field (CRF) (Krähenbühl and Koltun, 2011) as a post-processing step. The energy function is modeled as:

$$E(\mathbf{y}) = \sum_i \psi_u(y_i) + \sum_{i<j} \psi_p(y_i, y_j),$$

where $y_i$ represents the label assignment for pixel $i$. The unary potential $\psi_u(y_i) = -\log P(y_i|\mathbf{x})$ is derived from the smoothed CAM probabilities. The pairwise potential $\psi_p$ enforces smoothness using Gaussian kernels:

$$\psi_p(y_i, y_j) = \mu(y_i, y_j) \left[ w_1 \exp\left( -\frac{|p_i - p_j|^2}{2\sigma_\alpha^2} \right) + w_2 \exp\left( -\frac{|p_i - p_j|^2}{2\sigma_\beta^2} - \frac{|I_i - I_j|^2}{2\sigma_\gamma^2} \right) \right],$$

where $p_i$ and $I_i$ denote the spatial position and color intensity vectors. Corresponding to our implementation, we use a smoothness kernel with $\sigma_\alpha = 15$ and weight $w_1 = 30$, and an appearance kernel with $\sigma_\beta = 10, \sigma_\gamma = 20$ and weight $w_2 = 50$. Inference is performed for 5 iterations to produce the final segmentation mask.

## 3. Experments and Results

### 3.1. Experimental Settings

**Dataset.** We evaluate our method on the BCSS-WSSS dataset (Han et al., 2021a), which contains four tissue classes: Tumor (TUM), Stroma (STR), Lymphocyte (LYM), and Necrosis (NEC). All images are resized to $224 \times 224$ pixels.

**Implementation Details.** Our framework employs two frozen foundation models: CONCH ViT-B16 (Lu et al., 2024) (86M parameters) for semantic features and SegFormer MiT-B1

(Xie et al., 2021) (13M parameters) for structural guidance. Only lightweight adapters (6.3M parameters, 3.7% of the total model parameters) are trainable. We initialize class prototypes using detailed pathology text descriptions encoded by the frozen CONCH text encoder (Table 1).

Table 1: Text descriptions for prototype initialization

| Class | Description |
|-------|-------------|
| **TUM** | Tumor regions consist of malignant epithelial cells characterized by pleomorphic, hyperchromatic nuclei, loss of glandular structure, and frequent mitotic figures, representing the core cancerous lesions in breast tissue. |
| **STR** | Stroma comprises the surrounding connective tissue, including fibroblasts and collagen fibers, often appearing as pink fibrous structures and sometimes showing desmoplastic reactions to tumor invasion. |
| **LYM** | Lymphocyte regions contain dense clusters of small immune cells with dark, round nuclei and minimal cytoplasm, reflecting the host immune response within the tumor microenvironment. |
| **NEC** | Necrosis represents areas of dead or dying tissue with pale, structureless eosinophilic regions, nuclear debris, and "ghost" cell remnants, typically associated with aggressive tumor growth and poor vascularization. |

**Training.** We train for 2 epochs using the AdamW optimizer with learning rate $2 \times 10^{-5}$, weight decay 0.001, and batch size 10. The total loss combines three components: BCE classification loss ($\lambda_{\mathrm{cls}} = 1.0$), structural distillation loss ($\lambda_{\mathrm{struct}} = 1.5$), and InfoNCE contrastive loss ($\lambda_{\mathrm{contrast}} = 0.2$). We use a memory bank of 2048 negative samples for contrastive learning. Distillation is performed only at layer 2 of the feature hierarchy.

**Inference.** We apply test-time augmentation with 6 augmentations (horizontal flip $\times$ brightness scaling $\{0.9, 1.0, 1.1\}$) and average the resulting CAMs. Background probability is estimated as $(1 - \max_c \mathbf{M}_c)^{10}$, and Dense CRF post-processing is applied to refine boundaries. Segmentation quality is quantified using mean Intersection-over-Union (mIoU) and mean Dice coefficient.

**Baselines.** We benchmark against leading WSSS methods: TPRO (Zhang et al., 2023), MLPS (Han et al., 2021b), Proto2Seg (Pan et al., 2023), and PBIP (Tang et al., 2025).

### 3.2. Results

**Quantitative Results.** Table 2 shows that our method achieves the best results on BCSS-WSSS, reaching 70.96 mIoU and 82.83 mDice, surpassing all prior approaches. It also achieves the highest IoU and Dice for tumor and necrosis—two of the most challenging classes due to their variable shapes and boundaries. Performance on stroma and lymphocytes remains competitive and on par with or better than existing methods. Overall, the results demonstrate consistent gains across most tissue types.

**Qualitative Results.** The quantitative results in Figure 2 suggest that the model identifies tissue regions more accurately than the comparison methods. Higher IoU and Dice for tumor, stroma, and necrosis indicate improved coverage of these structures and fewer boundary inconsistencies. The gains across several classes further imply that the learned

| Method | Metrics (%) | | Per-class IoU (%) | | | | Per-class Dice (%) | | | |
|---|---|---|---|---|---|---|---|---|---|---|
| | mIoU | mDice | TUM | STR | LYM | NEC | TUM | STR | LYM | NEC |
| **TPRO** | 65.54 | 78.93 | 77.29 | 66.83 | 56.81 | 61.23 | 87.19 | 80.12 | 72.46 | 75.95 |
| **MLPS** | 61.58 | 75.95 | 72.98 | 62.58 | 52.03 | 58.73 | 84.38 | 76.99 | 68.45 | 74.00 |
| **Proto2Seg** | 57.42 | 72.24 | 63.25 | 58.28 | 53.27 | 54.89 | 77.49 | 73.64 | 67.78 | 70.08 |
| **PBIP** | 69.42 | 81.84 | 77.92 | 64.68 | **65.40** | 69.69 | 87.59 | 78.56 | **79.08** | 82.14 |
| **Ours** | **70.96** | **82.83** | **81.59** | **67.99** | 62.70 | **71.56** | **89.86** | **80.95** | 77.08 | **83.42** |

Table 2: Segmentation results on **BCSS-WSSS**

features work more consistently across different tissue types, leading to more stable segmentation performance.

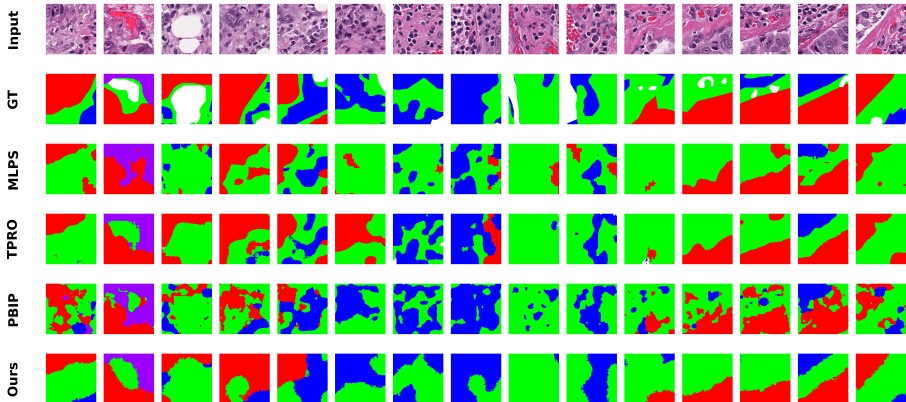

Figure 2: Qualitative results on BCSS-WSSS test patches. GT denotes ground truth segmentation

### 3.3. Ablation Study

**Ablation on Relational Structural Distillation.** We study the contribution of the relational structural distillation module by comparing the full model with a variant trained without it. As shown in Table 3, removing distillation reduces overall performance, indicating its importance. Distillation improves both mIoU and mDice, with the largest gains observed for tumor and necrosis, which contain complex and irregular boundaries. Stroma also benefits, though to a smaller extent. Lymphocyte performance is slightly higher without distillation, likely because these small, sparse regions are more sensitive to smoothing.

To further understand these effects, Figure 3 shows feature maps from three example images, comparing models trained with and without distillation, where distillation is applied at Stage 3. Without distillation, the feature maps tend to mix foreground and background. As seen when compared with the ground-truth masks, the non-distilled model often misses parts of the relevant structures. In contrast, with distillation, the feature maps become

Table 3: Ablation of Relational structural distillation on **BCSS-WSSS**.

| Method | Metrics (%) | | Per-class IoU (%) | | | | Per-class Dice (%) | | | |
|---|---|---|---|---|---|---|---|---|---|---|
| | mIoU | mDice | TUM | STR | LYM | NEC | TUM | STR | LYM | NEC |
| **w distill.** | **70.96** | **82.83** | **81.59** | **67.99** | 62.70 | **71.56** | **89.86** | **80.95** | 77.08 | **83.42** |
| **w/t distill.** | 70.40 | 82.47 | 80.82 | 67.47 | **63.58** | 69.73 | 89.39 | 80.57 | **77.74** | 82.17 |

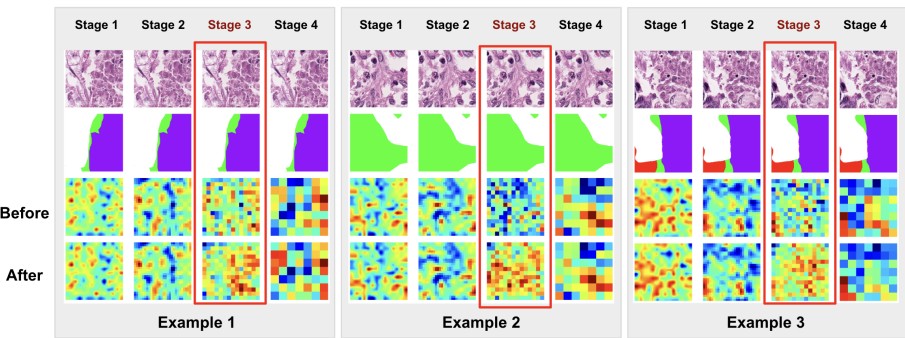

Figure 3: Qualitative comparison of relational structural distillation on three representative examples. For each example, the top row shows the original histopathology image, the second row shows the ground-truth segmentation mask, and the bottom rows visualize the average feature activation maps at each stage. **"Before"** denotes the model trained without relational structural distillation, while **"After"** denotes the full model with distillation applied (only at Stage 3).

cleaner and more spatially organized: foreground regions are activated more completely and accurately. This produces more interpretable spatial patterns and enables the model to better capture the true structure indicated in the masks. Overall, both the quantitative metrics and visual evidence show that relational structural distillation yields clearer feature representations and leads to more consistent segmentation results.

## 4. Conclusion

In this work, we explored a unified framework for weakly supervised histopathology segmentation, aiming to bridge the gap between global semantic understanding and local structural precision. By distilling spatial priors from a SegFormer teacher into a CONCH student via lightweight adapters, we demonstrated that fine-grained structural details can be transferred to foundation models. Furthermore, our use of text-guided prototype initialization suggests that integrating pathology descriptions enhances class discriminability for complex tissues. Experiments on the BCSS-WSSS dataset indicate that our approach yields robust performance with high parameter efficiency, offering a promising direction for resource-constrained, label-efficient medical image analysis.

## Acknowledgments

This work was supported in part by the National Institutes of Health (NIH) under Grant 5R01DK134055-02.

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
