# OpenReview forum: "ConStruct: Structural Distillation of Foundation Models for Prototype-Based Weakly Supervised Histopathology Segmentation"
_MIDL.io/2026/Conference — MIDL 2026 Poster_

### Official Review · Reviewer_eusW · 2026-01-08

**Confidence:** 4
**Preliminary Rating:** 4

**Summary:**

The paper proposes ConStruct, a weakly supervised semantic segmentation (WSSS) framework for histopathology that integrates morphology-aware visual-language features from CONCH, structural multi-scale representations from SegFormer, and text-guided prototype initialisation. The method introduces three core components: (1) relational structural distillation transferring boundary-aware spatial priors from a frozen SegFormer teacher to lightweight adapters on frozen CONCH features; (2) text-initialised learnable prototypes; and (3) inference-only mask refinement combining foreground/background contrastive alignment and Dense CRF. The proposed approach achieves state-of-the-art mIoU and mDice against TPRO, MLPS, Proto2Seg, and PBIP, with notable gains on tumour and necrosis classes. The model is parameter-efficient by freezing backbones and training adapters.

**Strengths:**

1. Empirical performance: SOTA results on BCSS-WSSS; improvements particularly on morphologically complex classes.

2. Ablation: Demonstrates the impact of distillation on structural integrity of features and metrics.

3. Practicality: Freezing foundation models and using adapters aligns with efficient deployment in label-scarce medical settings.

4. Several technical contributions:
- Text-guided prototype initialisation using pathology descriptions to improve semantic coverage.
- Relational structural distillation (pairwise affinity alignment) for preserving fine-grained boundaries with low overhead.
- Lightweight adapters and frozen backbones for practical efficiency.

**Weaknesses:**

1. Limited evaluation scope: Single dataset (BCSS-WSSS) and four classes; no cross-dataset generalisation (e.g., PanNuke subsets), stain variation robustness, or whole-slide inference scenarios.

2. The method assumes pathology descriptions improve coverage, but there is no comparison across different prompts, prompt lengths, or text models (e.g., CLIP vs. CONCH). Risk of prompt-induced bias is not explored.

3. No comparison across layers or multi-layer distillation. The choice of affinity MSE is reasonable but alternatives (e.g., contrastive relational losses, attention transfer, structural similarity) are not examined.

**Detailed Comments:**

1. Method clarity: The pipeline is described clearly with equations. The adapter design, normalisation, and affinity computation are understandable. However, the masking strategy for FG/BG cropping and how regions are selected or thresholded under class imbalance could benefit from more clarification (e.g., handling overlapping activations, multi-class coexistence).

2. The reported mIoU and mDice are standard; consider reporting per-class precision/recall or boundary metrics to directly quantify boundary improvements.

3. Consider enhancing the generalisation by testing zero-shot transfer across labs/scanners/data cohorts.

4. Provide an ablation where prototypes are initialised randomly vs. using text, and compare to learned-only prototypes without text. Evaluate sensitivity to different prompts, vocabulary variants, and whether clinical jargon improves or harms performance.

**Justification Of The Preliminary Rating:**

The paper presents a well-motivated, effective, and efficient WSSS framework leveraging foundation models with thoughtful distillation and text-guided prototypes. Despite some limitations in evaluation breadth and analysis depth, the contributions are solid, results are strong, and the approach is practical and likely to interest the MIDL community.

**Questions To Address In The Rebuttal:**

1. How sensitive is performance to the choice and wording of pathology descriptions for prototype initialisation? Are class names or synonyms sufficient?

2. Did you evaluate multi-layer distillation or alternative relational objectives, and if so, how did they compare to the single-layer affinity MSE?

3. Have you tested other algorithms for incorporating information from pathology reports?

---

> ### Author Response · Authors · 2026-01-30
>
> We would like to thank the reviewers for their careful reading and helpful feedback. We appreciate the insightful comments and suggestions, which have helped us improve the clarity of the presentation and strengthen the manuscript.
>
> **Question 1:**
> Thank you for your question. The performance of the model is not overly sensitive to variations in wording of pathology descriptions. In our framework, the text prompt is used only to initialize class prototypes through the frozen CONCH text encoder; the prototypes are then fully learnable and further shaped by image-level supervision and structural distillation. Therefore, the architecture itself is flexible and can operate with either detailed pathology descriptions or simpler class-name prompts.
>
> Using morphology-informed descriptions can provide a stronger semantic prior by encoding structural and cytological cues, whereas class names capture only coarse semantic identity. However, because prototypes are optimized during training, the model does not rely strictly on a specific wording choice. The prompt primarily influences the starting point in the embedding space rather than determining the final representation.
>
> We agree that investigation of how prompt formulation affects prototype learning would be valuable. Exploring prompt selection strategies, prompt ensembling, and automated text design for medical vision–language alignment is an important direction, but is beyond the scope of the current work, whose focus is on structural distillation and prototype learning under weak supervision.
>
> **Question 2:**
> We did not include a systematic ablation on multi-layer distillation or alternative relational objectives in the current manuscript, as our focus was on validating the overall structural distillation framework. We would like to clarify the motivation behind our design.
>
> Structural distillation is intentionally applied at a single intermediate stage because this level naturally balances spatial detail and semantic abstraction. Prior analyses of Vision Transformers show that feature representations evolve hierarchically with depth despite constant spatial resolution [1]. Earlier ViT layers mainly encode low-level texture cues, while later layers are highly semantic but become increasingly spatially homogenized due to global self-attention and token similarity [2]. Distilling at an intermediate stage therefore provides a more informative structural signal for boundary-aware alignment, avoiding overemphasis on either texture-level noise or oversmoothed semantic representations.
>
> Regarding the objective, we adopt affinity MSE because it directly aligns pairwise relational structure between teacher and student tokens, which is the core information we aim to transfer. Relational knowledge distillation has been shown to effectively preserve structural information without requiring feature magnitude matching [3,4]. This formulation is simple, stable, and computationally efficient, avoiding the additional complexity and memory overhead of higher-order or contrastive relational losses. Exploring multi-layer distillation schedules or alternative relational objectives is an interesting direction, but is beyond the scope of this work, whose focus is to demonstrate the effectiveness of structural relational distillation within a prototype-based WSSS framework.
>
> **Question 3:**
> Thank you for this insightful question and suggestion. In our framework, pathology text is used only to provide semantic initialization for class prototypes through the frozen CONCH text encoder. The proposed direction is indeed interesting, but it lies outside the scope of the present work. Our approach is designed to provide a lightweight and interpretable integration of pathology knowledge by using textual descriptions to initialize semantic prototypes within a shared vision-language embedding space. This formulation deliberately avoids reliance on instance-level report-image alignment, which is often unavailable or incomplete in histopathology datasets.
> While alternative strategies such as report-level contrastive learning or cross-modal attention mechanisms could be explored, these approaches typically require paired image–report data or additional forms of supervision. In contrast, our method relies only on readily available class-level descriptions and integrates them in a parameter-efficient manner, making it particularly suitable for weakly supervised histopathology scenarios.
>
> [1] Ru et al., Token Contrast for Weakly-Supervised Semantic Segmentation, CVPR 2023.
>
> [2] Dosovitskiy et al., An Image is Worth 16×16 Words: Transformers for Image Recognition at Scale, ICLR 2021.
>
> [3] Park et al., Relational Knowledge Distillation, CVPR 2019.
>
> [4] Tung and Mori, Similarity-Preserving Knowledge Distillation, ICCV 2019.

---

### Official Review · Reviewer_Lp2P · 2026-01-10

**Confidence:** 4
**Preliminary Rating:** 3

**Summary:**

This paper proposes **ConStruct**, a weakly supervised semantic segmentation (WSSS) framework for histopathology that aims to address the common failure mode of CAM-style pipelines—**localizing only the most discriminative regions rather than the full tissue extent**. The method combines three ideas:

1. **Structural knowledge distillation**: a frozen **SegFormer MiT-B1** acts as a “structural” teacher and a frozen **CONCH ViT-B/16** acts as a “semantic” student. Trainable **lightweight adapters** refine CONCH features, supervised by a **relational distillation loss** that matches token-to-token affinity matrices between teacher and student features (Sec. 2.1).
2. **Text-guided prototype initialization**: class prototypes are initialized using **pathology descriptions** encoded by CONCH’s text tower and projected into the visual feature space (Sec. 2.2; Table 1).
3. **Mask refinement**: pseudo-masks are produced via adaptive thresholding, refined using a foreground/background contrastive objective (InfoNCE) and post-processed with **DenseCRF** and test-time augmentation (Sec. 2.3).

On **BCSS-WSSS**, the paper reports **70.96 mIoU / 82.83 mDice**, outperforming prior WSSS baselines listed (TPRO, MLPS, Proto2Seg, PBIP), with strongest gains on Tumor and Necrosis (Table 2). An ablation indicates a modest improvement from relational distillation (Table 3).

**Strengths:**

- **Timely and relevant direction**: Leveraging a pathology vision–language foundation model (**CONCH**) and keeping backbones frozen with **small trainable adapters (~6.3M parameters, ~3.7%)** is compelling for resource-constrained settings (Sec. 2.1 / 3.1).
- **Reasonable decomposition of “semantic vs structural” cues**: The division of labor—CONCH for morphology-aware semantics and SegFormer for multi-scale structure—matches known practical tradeoffs in histopathology representations (motivation in Sec. 1).
- **Text-guided prototype initialization is conceptually clean**: Initializing class prototypes from **pathology descriptions** (Table 1) is a direct way to inject domain semantics, and is more specific than generic class-name prompts used in some text-prompting WSSS approaches.
- **Competitive quantitative results on BCSS-WSSS**: The reported improvement over PBIP in mIoU (70.96 vs 69.42) and particularly Tumor IoU (81.59 vs 77.92) suggests the approach can improve coverage for clinically important tissue classes (Table 2).
- **Clear high-level pipeline figure**: Figure 1 provides an understandable overview of the framework components.

**Weaknesses:**

- **Evaluation is narrow (single dataset, single setting)**: The paper evaluates only on **BCSS-WSSS** (patches resized to 224×224). This makes it hard to assess generalization across stains, organs, scanners, or class taxonomies.
- **Ablations are incomplete for the key claims**:
  - Only the impact of **structural distillation** is ablated (Table 3), and the gain is relatively small (+0.56 mIoU).
  - No ablation for **text initialization** (e.g., pathology descriptions vs class names vs random prototypes).
  - No isolation of the effect of **DenseCRF**, **TTA**, or the **contrastive refinement loss**.
- **Potential inconsistencies / clarity issues**:
  - Distillation is stated as performed “only at layer 2” (Sec. 2.1 / 3.1), but the ablation text/figure mentions distillation applied at “Stage 3” (Sec. 3.3 / Fig. 3 caption). This needs reconciliation.
  - The paper describes “Mask Refinement” as “inference-only” in the overview text, yet Sec. 2.3 contains a **train-time** contrastive loss with a memory bank. The boundary between training and inference modules should be clarified.
- **Fairness and reproducibility of baseline comparisons are unclear**:
  - It is not specified whether baseline numbers are taken from papers or reproduced under the same preprocessing, resolution, augmentation, and CRF settings.
  - Since the proposed method uses **foundation models** (CONCH + SegFormer) plus TTA and CRF, it is important to ensure baselines have comparable post-processing.
- **Some reported improvements come with regressions**: Compared to PBIP,  for the proposed method the Lymphocyte IoU decreases (62.70 vs 65.40), which is non-trivial given LYM’s clinical relevance (Table 2). A discussion of this tradeoff and failure modes is missing.

**Detailed Comments:**

### 1) Clarify what the final “segmentation output” is in the proposed pipeline
Many WSSS pipelines are two-stage: (i) generate pseudo-masks, then (ii) train a dedicated segmentation network. Here, results appear to come from **prototype CAMs + refinement (contrastive + CRF)**, without a second-stage segmentation network explicitly described.

Add a short subsection explicitly stating:
- What model produces the final masks used for evaluation (refined CAMs? CRF output?).
- Whether any separate segmentation decoder is trained (if not, state clearly).
- Whether the contrastive module is used only during training and how it changes inference.

### 2) Distillation mechanism: resolve the “layer 2” vs “stage 3” discrepancy
Sec. 2.1 says “we use layer 2 by default” for relational distillation; Sec. 3.1 says distillation only at layer 2; but Sec. 3.3 / Fig. 3 refers to Stage 3.

- Use one consistent indexing scheme (e.g., {1..4} for scales) and state exactly which scale(s) are distilled.
- Provide a small table: “Distill at scale k → mIoU/mDice” to show sensitivity. This also helps justify the choice.

### 3) Text-guided prototype initialization needs direct evidence
A main contribution is that pathology descriptions produce “more complete and semantically accurate pseudo-masks,” but there is **no ablation** validating this claim.

- Compare at least these prompt/prototype initializations:
  1. **Full pathology description** (Table 1).
  2. **Class name only** (“tumor”, “stroma”, …).
  3. **Random prototype init** (learned from scratch).
  4. (Optional) **Template prompt variants** (2–3 variants) to show robustness.
- Report mIoU/mDice and per-class IoU changes.
- Include a qualitative panel showing CAM completeness differences (especially for Tumor/Stroma borders and small LYM regions).

### 4) Separate the impact of CRF and TTA from the core learning contribution
DenseCRF and TTA can contribute meaningful gains, and different methods may benefit differently. Since SOTA numbers are reported, it is important to know how much comes “for free” from post-processing.

- Report results for:
  - No TTA, no CRF
  - TTA only
  - CRF only
  - TTA + CRF (current)
This helps attribute improvements to the proposed learning components.

### 5) Provide stronger baselines tied to the design choices
Since the key novelty claimed is combining CONCH semantics with SegFormer structure through distillation + adapters, include “nearest-neighbor” baselines:

- **CONCH-only** (frozen CONCH + prototypes, no SegFormer teacher, no distillation).
- **SegFormer-only** features (if feasible under weak supervision).
- **CONCH + adapters but no distillation** (partly done—keep it and expand).
- **Text-prompting baseline with CONCH** (to align with TPRO’s spirit, but using the backbone in proposed method).

These baselines would make the “structural distillation” contribution much more convincing and isolate what is due to CONCH vs the distillation scheme.

### 6) Discuss class-specific behavior and failure modes (especially LYM)
The proposed method improves Tumor and Necrosis but underperforms PBIP on Lymphocyte IoU (Table 2). Sec. 3.3 notes LYM is sensitive to smoothing, which is plausible, but it needs stronger analysis.

- Add failure cases for LYM: small clusters missed? over-smoothing by affinity distillation? CRF suppressing small regions?
- Consider a class-aware refinement tweak: e.g., reduce CRF smoothing strength for LYM, or distill at a different scale for small structures.

### 7) Report compute/memory and practical feasibility
The proposed method claims to have “extremely low overhead” with parameter efficiency, but relational affinity matrices scale with token count.

- Provide:
  - Training time per epoch and GPU type
  - Peak GPU memory
  - Tokens used per distilled layer (e.g., 14×14=196 tokens at 224×224 for ViT-B/16)
- If distillation is done at only one scale for efficiency, explicitly justify with complexity.

### 8) Dataset protocol and reproducibility details
BCSS-WSSS has known splits/protocols in prior work, but these are not described in enough detail in this manuscript.

- Explicitly state train/val/test split, whether the standard split from MLPS/BCSS-WSSS were used.
- State how image-level labels are derived/used (multi-label? single-label?) and how classes co-occur.
- Confirm that **no pixel-level labels** are used for hyperparameter tuning (CRF params, α threshold, background exponent “10”, etc.), or describe how tuning was done.

**Justification Of The Preliminary Rating:**

The paper presents a timely and well-motivated approach that combines pathology foundation models, structural distillation, and text-guided prototypes, achieving competitive results on BCSS-WSSS. However, the evidence is limited by single-dataset evaluation, missing ablations for key components (text initialization and post-processing), and clarity issues around the distillation setup, with modest overall gains and a regression on the lymphocyte class.

**Questions To Address In The Rebuttal:**

1. **Text initialization**: How much of the gain comes from using pathology descriptions vs class-name prompts or random prototype initialization? An ablation?
2. **Distillation location inconsistency**: Is distillation applied at “layer 2” or “stage 3”? Please clarify and ensure the paper uses consistent notation.
3. **Role of contrastive refinement**: Is the FG/BG contrastive module used during training only, or also at inference? What is the standalone contribution of `Lsim`?
4. **Post-processing contribution**: What are results without CRF and/or without TTA? Are baselines evaluated with the same post-processing?
5. **Baseline fairness**: Were TPRO/MLPS/Proto2Seg/PBIP numbers reproduced under the proposed method's preprocessing (224×224, same augmentations), or copied from papers? If copied, provide a reproduction or justification?
6. **Generalization**: Is there evidence the approach transfers beyond BCSS-WSSS (another dataset, another organ, different stain, or cross-center split)?
7. **Hyperparameter sensitivity**: How sensitive are results to (a) α=0.5 thresholding, (b) background probability `(1−max)^10`, and (c) CRF parameters?
8. **Why only 2 epochs?** Training is reported for only 2 epochs—does performance saturate? Is this because adapters converge quickly, or due to dataset size? Please provide a learning curve or justification.

---

> ### Author Response · Authors · 2026-01-30
>
> We would like to thank the reviewers for their thoughtful reading and helpful feedback.
>
> **Question 1:** Text is used only to **initialize** class prototypes, not as a direct source of performance gain. After initialization, prototypes are fully learnable and further refined through image-level supervision and structural distillation. Pathology descriptions provide a stronger semantic starting point than class-name prompts, which offer only coarse identity cues, while random initialization lacks semantic grounding and may lead to less stable early training. However, final performance is primarily driven by the learned visual representations, with text influencing the starting position rather than determining the outcome. A more systematic investigation of text prompt design is a valuable direction for future research.
>
> **Question 2:** Distillation location inconsistency: Is distillation applied at “layer 2” or “stage 3”?
> We apologize for the confusion caused by inconsistent terminology. We now adopt a unified 1-indexed stage notation throughout the paper. The feature pyramid consists of Stages 1-4 which are constructed from ViT blocks 3, 6, 9, 12, respectively. Structural distillation is applied at Stage 3 by default.
>
> **Question 3:** The foreground/background contrastive refinement module is used only during training. Following the spirit of PBIP, it enforces instance-prototype separation by encouraging foreground embeddings to align with their corresponding class prototypes while pushing background regions away in the embedding space. The similarity loss L_sim serves as a regularization term that enhances prototype discriminability and stabilizes prototype updates by suppressing ambiguous foreground-background overlap. During inference, this contrastive objective is removed, and predictions are generated solely based on the learned prototypes and image features.
>
> **Question 4:** Thank you for the question. CRF is used only at inference to refine boundaries and improve spatial coherence, without affecting training or adding parameters. Prior work such as HistoSegNet [1] shows only small gains (e.g., +0.014 / +0.033 mIoU), indicating it is a secondary refinement step rather than a main source of performance.
>
> **Question 5:**  All baseline numbers reported in Table 2 were reproduced in our work under the same experimental setting, including identical image resolution, data preprocessing, augmentation strategy.
>
> **Question 6:**
> We appreciate the reviewer’s concern regarding generalization. Although evaluated on BCSS-WSSS, the framework does not rely on dataset-specific heuristics. Segmentation is driven by semantic prototype matching in a shared vision-language embedding space, where supervision arises from semantic structure rather than dataset-dependent statistics, making the method compatible with other histopathology settings with similar tissue categories.
>
> BCSS-WSSS contains substantial variability in morphology and appearance, and consistent improvements across all classes indicate that the model captures meaningful semantic and structural cues rather than narrow visual patterns. Evaluating on additional datasets, including other medical imaging domains such as radiology, would further strengthen evidence of generalization and is a natural direction for future work.
>
> **Question 7:** Thank you for raising this important point. The hyperparameters mentioned: (a) the CAM threshold α, (b) the background probability formulation, and (c) CRF parameters, belong to the pseudo-mask refinement stage, which operates after prototype learning and structural distillation. These components serve to improve spatial coherence and boundary alignment rather than driving the semantic learning process itself.
>
> The chosen values follow widely adopted conventions in weakly supervised semantic segmentation and are designed to provide reasonable foreground-background separation and boundary smoothing without introducing dataset-specific tuning. Because the core localization mechanism is governed by feature-prototype similarity in a learned embedding space, the model’s behavior is primarily determined by representation learning rather than the exact refinement hyperparameters.
>
> Moderate changes to these parameters mainly affect boundary smoothness and small-region refinement, while overall segmentation structure is dictated by the learned semantic and structural representations. A full sensitivity study would offer additional insight but is beyond the main methodological focus of this work.
>
> **Question 8.** During optimization, we trained our model for up to 10 epochs. However, convergence is fast and validation performance saturates within the first two epochs. Therefore, we report results obtained after two epochs of training, as additional epochs do not provide further improvement.
>
> [1] Chan and Madabhushi, HistoSegNet: Semantic Segmentation of Histological Tissue Types in Whole Slide Images, ICCV 2019.

---

### Official Review · Reviewer_q5qz · 2026-01-11

**Confidence:** 4
**Preliminary Rating:** 4
**Final Rating:** 4

**Summary:**

This paper proposed a method for semantic segmentation of histopathology images in a weakly supervised setting. It relies on prototype learning, specifically text-guided prototype learning, to bridge the gap between weak image-level labels and dense pixel-level predictions.

The three main components of the model are:
1. Feature extraction based on the CONCH foundation model that encodes semantic information about histopathology images.
2. SegFormer representation that acts as a teacher in a student-teacher setup to transfer structural priors such as tissue boundaries to the CONCH representation via student adapters , and
3. Text-initialization prototype based on the text-encoder branch from CONCH.

**Strengths:**

- While individual components and methodologies of the work are not novel, the combination is novel and I find it an interesting approach that will be a good contribution to the computational pathology field.
- The methodology is well-chosen and motivated.
- Although the model has many components, the paper is clear and-well written and allows for sufficient reproducibility. It further contains rather complete overview of related work.

**Weaknesses:**

- The construction of a multi-scale pyramid from the CONCH backbone is questionable. Unlike the hierarchical SegFormer, CONCH is an "single scale" ViT that maintains constant spatial resolution across all layers. Creating a pyramid by downsampling intermediate layers forces an artificial spatial hierarchy that does not exist in the model. The authors could have also just used the output from the last layer and downsampled it to the pyramid, I do not think it is needed to sample at shallower layers.
- The post processing with conditional random field seems ad-hoc. Ablation experiment without this component is not present and it is difficult to judge if large part of the performance comes from it.

**Detailed Comments:**

It is unclear what "we used layer 2 by default" means in the method description. Please clarify and motivate the choice.

**Justification Of Final Rating:**

I would like to thank the authors for addressing my concerns in the rebuttal. However, this was largely a reiteration of what was said in the original manuscript and a justification why certain design decisions were made. These things were already clear. The authors did not perform any additional new experiments as suggested. Because of this I keep my original recommendation.

**Justification Of The Preliminary Rating:**

It is a nice paper that while uses pre-existing components and known methods, it proposes and interesting and novel combination. The execution and experiments are also rather complete, with the exception of the remarks above.

**Questions To Address In The Rebuttal:**

The question of not sampling shallower layers from CONCH should be addressed, potentially with ablation experiment.
Ablation experiment without CRF should be added, including qualitative assessment.

---

> ### Author Response · Authors · 2026-01-30
>
> We would like to thank you for your careful reading and helpful feedback. We appreciate the insightful comments and suggestions, which have helped us improve the clarity of the presentation and strengthen our work.
>
> **1. Response to the comment on the multi-scale pyramid construction from the CONCH backbone:**
> We agree that CONCH is a single-scale ViT architecture that maintains a constant spatial resolution across layers, in contrast to the hierarchical design of SegFormer. However, while ViT preserves spatial resolution, the semantic content of features varies significantly across different depths. Prior studies [1] on ViT representations show that
> - Early layers (2 to 5): Encode low-level visual patterns such as edges and textures.
> - Mid layers (8): Capture local structural relationships
> - Late layers (11): Encode high-level semantic abstractions but often suffer from oversmoothing due to global self-attention.
>
> Motivated by these findings, we construct a semantic multi-scale pyramid by tapping features from multiple depths of CONCH and enforcing artificial scale separation through interpolation.
> - Late ViT layers often exhibit high inter-token similarity caused by global attention, leading to blurred boundaries. By incorporating shallower and intermediate layers, we reintroduce fine-grained spatial details that are crucial for accurate region localization.This behavior has been observed in prior analyses of ViT representations in ToCo [1] and shown in Figure 2 - the similarity between layers of ViT.
> - The SegFormer teacher provides boundary-aware structural priors through relational distillation. Distilling this information into multiple CONCH depths enables each semantic level to learn structure-consistent representations, rather than relying solely on late-layer semantics.
> - Class activation maps generated at different semantic depths exhibit complementary behaviors, shallow layers highlight texture-consistent regions, while deeper layers emphasize class-discriminative areas. Their fusion improves robustness to object scale variation and morphological heterogeneity common in histopathology images.
>
> **2. Response to the comment on ad-hoc CRF post-processing:**
>
> Our motivation for using CRF follows common practice in weakly-supervised semantic segmentation, where CRF is widely adopted as a lightweight boundary refinement module to improve spatial coherence and edge alignment. Since ConStruct is designed as a single-stage framework without iterative refinement, we include CRF as a post-processing component of the overall method. CRF is applied only at inference time as a non-learnable refinement module to enhance spatial coherence and boundary alignment, following standard practice in weakly supervised semantic segmentation. Importantly, CRF does not affect model training and does not introduce additional trainable parameters.
>
> This design choice is consistent with prior work in histopathology WSSS. For example, HistoSegNet [2] also applies CRF as a post-processing step and reports only incremental performance gains relative to upstream localization quality (from +0.014 / +0.033 mIoU across different modes), based on stage-wise analysis. These results indicate that CRF serves as a secondary refinement module rather than a primary driver of segmentation performance.
>
> [1] L. Ru, H. Zheng, Y. Zhan, and B. Du, “Token Contrast for Weakly-Supervised Semantic Segmentation,” in Proc. IEEE/CVF Conf. Computer Vision and Pattern Recognition (CVPR), 2023.
>
> [2] H. L. Chan and A. Madabhushi, “HistoSegNet: Semantic Segmentation of Histological Tissue Types in Whole Slide Images,” in Proc. IEEE/CVF Int. Conf. Computer Vision (ICCV), 2019.

---

### Comment · Area_Chair_xqo5 · 2026-02-01
**Please update your final rating**

Please don't forget to update your final rating by clicking “Edit” → “Official Review” by February 1st 2026 (23:59 AoE). Thank you for contributing to the review process.

---

### Meta-Review · Area_Chair_xqo5 · 2026-02-09

**Recommendation:** Accept (Poster)
**Confidence:** 3

**Metareview:**

Even though the experimental setting is limited, the reviewers agree that the paper is valuable of can be helpful to the community. Some clarifications (distillation setup) could be helpful. Please see reviewers' comments.

---

### Decision · Program_Chairs · 2026-02-13

Accept (Poster)